# Steganography and Steganalysis in Voice over IP: A Review

**DOI:** 10.3390/s21041032

**Published:** 2021-02-03

**Authors:** Zhijun Wu, Junjun Guo, Chenlei Zhang, Changliang Li

**Affiliations:** School of Electronics & Information & Automation, Civil Aviation University of China, Tianjin 300300, China; 2019022120@cauc.edu.cn (J.G.); 2018022093@cauc.edu.cn (C.Z.); 2017021010@cauc.edu.cn (C.L.)

**Keywords:** VoIP, steganography, steganalysis, protocol, payload

## Abstract

The rapid advance and popularization of VoIP (Voice over IP) has also brought security issues. VoIP-based secure voice communication has two sides: first, for legitimate users, the secret voice can be embedded in the carrier and transmitted safely in the channel to prevent privacy leakage and ensure data security; second, for illegal users, the use of VoIP Voice communication hides and transmits illegal information, leading to security incidents. Therefore, in recent years, steganography and steganography analysis based on VoIP have gradually become research hotspots in the field of information security. Steganography and steganalysis based on VoIP can be divided into two categories, depending on where the secret information is embedded: steganography and steganalysis based on voice payload or protocol. The former mainly regards voice payload as the carrier, and steganography or steganalysis is performed with respect to the payload. It can be subdivided into steganography and steganalysis based on FBC (fixed codebook), LPC (linear prediction coefficient), and ACB (adaptive codebook). The latter uses various protocols as the carrier and performs steganography or steganalysis with respect to some fields of the protocol header and the timing of the voice packet. It can be divided into steganography and steganalysis based on the network layer, the transport layer, and the application layer. Recent research results of steganography and steganalysis based on protocol and voice payload are classified in this paper, and the paper also summarizes their characteristics, advantages, and disadvantages. The development direction of future research is analyzed. Therefore, this research can provide good help and guidance for researchers in related fields.

## 1. Introduction

Information hiding technology is also called steganography [1]. The principle is to hide secret information without being noticed by a third party by modifying redundant data in digital media or protocols, such that the carrier’s use attributes are not changed during transmission. By this means, a secret message can be embedded into cover objects and transmitted through public channels [2,3]. At present, it is widely used in transmission media such as voice, image, video, and text. Different types of carriers have distinctive information hiding algorithms. Because human organs are insensitive (for example, the ears are not very perceptive to subtle changes in sound [4]), people are not able to use their senses to discover the difference between the original carrier (the carrier that does not contain secret information) and the secret carrier (the carrier that contains secret information), and they are unable to discover the covert communication. Distinct from encryption technology, steganography technology provides a more reliable and safe method of information transmission by hiding the location and method of embedded information to make the information undetectable to third parties [5,6]. Encryption technology causes the transmitted cipher text to have an obvious sense of “violation”, which is more likely to arouse the alertness of attackers. Once discovered, the attacker can use various approaches and tools such as brute force cracking to destroy the cipher text, which greatly increases the risks with secret communication. In short, encryption technology hides the content of covert communication [7,8], while steganography technology hides the “behavior” of covert communication, so steganography technology is able to provide better concealment and security [9,10]. Table 1 compares the differences between steganography and encryption in five aspects [11].

Steganography is a new way to ensure information security, but it also has a double-edged sword effect, like many other things. On the one hand, it protects the safe and reliable transmission of private information and confidential information in political, financial, and other domains on public networks. On the other hand, it also provides opportunities for some criminals with improper or even malicious purposes. For example, steganography is used to hide the computer virus in various multimedia carriers to evade the review of firewalls and anti-virus software to carry out sabotage activities. It can be seen that the abuse of steganography technology will lead to the dissemination of illegal information that undermines state and social stability on the Internet and brings potential and destructive threats to the safety of people’s lives and property. Therefore, steganalysis technology, as a countermeasure against steganography, has drawn increasing attention from researchers [12].

Steganalysis is a confrontation technology of steganography. Its target is to discover the presence of secret information and even damage confidential communication. Steganalysis is a vital technology for resolving the issue of criminal use of steganography [13]. The improvement of steganalysis technology helps avoid the illicit appliance of steganography and can play a role in preventing the loss of private data, revealing illegal data, combating violence, preventing tragedies, and then ensuring public safety and social stability. Steganalysis not only has vital use value but also has significant literary importance. Steganalysis research can disclose the shortcomings of present steganography and estimate the safety of steganography. This is a useful technique for the development and improvement of message hiding methods.

VoIP (Voice over IP), also called IP telephony, is a method and group of technologies for the delivery of voice communications and multimedia sessions over Internet Protocol (IP) networks, such as the Internet. The system includes terminal equipment, gateways, gatekeepers, network management, etc. The traditional telephone network transmits voice in a circuit-switched manner. VoIP uses an IP packet switching network as the transmission platform to encode and compress analog signals, and then package the voice data following the TCP/IP standard and other special processing, so that it can be transmitted using the connectionless UDP protocol. After decoding and decompression processing, it is restored to the original voice signal, to achieve the purpose of transmitting voice through the Internet. The VoIP transmission procedure is presented in Figure 1 [14].

VoIP has been praised by an increasing number of people because of the widespread use of the network and its convenience and timeliness. At the same time, it has become a major transmission carrier for steganography. The specific reasons for this are as follows:A protocol stack with a multi-layer protocol can embed secret information at the network layer, transport layer, and application layer by modifying the protocol header and other methods to achieve the purpose of covert communication.More steganographic possibilities can be provided in the process of encoding and packaging voice data.Because VoIP expands the transmission path including the IP network and the telephone network, the data transmission is difficult to detect.VoIP has a huge data volume because of its extensive use, which can include embedded secret information, and it is difficult to detect.The communication is instantaneous, with few restrictive conditions, and secret data can be steganographically written anytime and anywhere, which enhances the operability and timeliness of steganography and also increases the difficulty of detection.

Figure 2 [15] illustrates the steganography and confrontation model of VoIP communication. Alice represents the sender who uses VoIP data information. The covert information is embedded into the original information through a steganography algorithm (Steg) before sending, and the original information becomes a carrier that carries the secret information. In the process of transmitting information on the communication channel, the third party Wendy will perform detection and interference (Dec/Jam) to determine whether the transmitted information contains covert information. The transmission of information may be interrupted if secret information is found. The VoIP steganography algorithm is to show the unknowable of secret information transmission. That is, to prevent third parties from discovering the possibility of clandestine information transmission. Bob is the receiver of the VoIP data message. The mystery message is extracted from the steganographic data through the (Extr) algorithm.

As we all know, there have been several reviews on VoIP steganography and steganalysis [15,16,17,18], but these published articles have not classified and discussed some specific steganography and steganalysis algorithms well. Therefore, the present article is motivated by the following factors:The published reviews do not have a detailed classification of VoIP steganography and steganalysis with respect to the difference of secret information embedding area and parameter attributes.An exhaustive introduction and performance comparison of steganalysis algorithms has not been carried out in the existing reviews.The need to summarize the latest VoIP steganography and steganalysis algorithms.

Therefore, it is necessary to classify, analyze, and summarize the published articles. Then, the advantages and disadvantages of the various algorithms can be summarized and compared in order to form a comprehensive review article to provide more accurate guidance for future researchers.

The aim of this paper is to show the advance of steganography and steganalysis based on VoIP. The novel contributions of this essay can be listed as follows:For the first time, the steganography and steganalysis algorithms are classified and summarized in detail with respect to both the embedding area and parameter attributes, simultaneously.The working principle of the VoIP steganographic transmission confrontation model is analyzed.The existing articles are classified and summarized, and the performance of steganography algorithms is compared in terms of imperceptibility, hidden capacity, and robustness. The performance of steganalysis algorithms is compared with respect to accuracy, applicability, and algorithm complexity.Newer steganography and steganalysis methods are summarized, and future development directions are proposed based on existing methods and challenges.

Furthermore, this article is compared with four other surveys. These are shown in Table 2.

This article first gives a detailed introduction to steganography, steganalysis, VoIP, etc., and explains the motivation and contribution of writing this article. The other parts of the essay are summarized as follows: Section 2 categorizes the existing steganography and steganalysis algorithms; Section 3 and Section 4 introduce steganography and steganalysis based on VoIP, respectively. In view of recent developments, Section 5 puts forward the future work and challenges of steganography and steganalysis based on VoIP; Section 6 summarizes the article.

## 2. Classification of Steganography and Steganalysis

Wojciech Mazurczyk [16] briefly analyzed and summarized the VoIP-based steganography and steganalysis algorithms available in 2003–2012, which is helpful for understanding the work at that stage. This article is a further improvement in terms of classification details and performance comparison with respect to [16], and the literature mentioned in [16] will not be considered. From the existing literature, VoIP steganography methods mainly have two research directions, according to the different steganographic areas: (1) steganography methods using the voice stream payload that is transmitted in real time by the VoIP system as the carrier; (2) network protocol steganography, which uses the network protocol involved in the VoIP transmission process as the carrier. Algebraic code excited linear prediction (ACELP) rule encoding is used in VoIP encoders, and the parameters obtained after encoding include linear predictive coefficient (LPC) parameters, fixed codebook (FCB) parameters, adaptive codebook (ACB) parameters [19], and gain parameters. Because the gain parameter redundancy is too small to embed more secret information, seriously reducing the embedding efficiency, it is unsuitable for steganography. The steganography methods focusing on the voice payload include the first three, or a mixture thereof.

Speech coding can realize the prediction of the short-term correlation of speech through the linear prediction algorithm, and complete the compression coding and transmission of the linear prediction coefficients and residual signals. A set of linear prediction coefficients (LPC) can be obtained after each analysis. The LPC must be converted into line spectrum pair frequency (LSF) parameters before encoding, and a set of LSF factors in every frame are quantized by split vector quantization (SVQ). The mean square error minimization criterion between the aggravated input speech and the weighted renewed speech is used to search the code vectors in the FCB. Each code vector contains two non-zero pulses, and the amplitude of each pulse is either positive or negative. The sample position of each frame is divided into five tracks, and each subframe takes two different track subsets, and each track subset contains two pulse positions. For a segment of the speech signal, its exact period cannot be determined. The pitch delay characterizing the period can be obtained through ACB search. The ACB search is performed on each subframe, including closed-loop pitch search and calculation of the adaptive codebook vector by interpolating the past excitation at the pitch delay. The adaptive codebook parameters are the pitch delay and pitch filter gain. The purpose of the ACB search is to obtain an optimal adaptive codebook index. In the search stage, the linear prediction residual expansion excitation simplifies the closed-loop search. Steganography and steganalysis algorithms are carried out in these processes. To improve the undetectability and robustness of the proposed method, the methods focus on classification according to parameter attributes. Because steganalysis is not universal, a certain steganalysis method can only detect a corresponding one. Therefore, the classification of steganalysis methods based on VoIP can be carried out following the steganography classification method. The specific classification of steganography and steganalysis methods are in Figure 3.

For the VoIP-based steganography methods, three indicators of imperceptibility, hidden capacity, and robustness are applied to evaluate the capability of the algorithms.


Imperceptibility. This means that the information hiding method uses the autocorrelation and statistical redundancy of the carrier data to embed the mystery message into the carrier without affecting the original quality of the carrier, making it difficult for third parties to discover. It can be evaluated in terms of spectrum, time domain, frequency domain, voice quality, etc.Hidden capacity. The hidden capacity is a measure of the hidden covert messages. It refers to the maximum number of bits that can be hidden in the carrier under the premise of satisfying the imperceptibility [1]. When applying information hiding methods in covert communication, to improve transmission efficiency, it is usually hoped that as many secret messages as possible can be hidden in each carrier. A hidden capacity that is too low often means low communication efficiency, and it is hard to fill the demands of covert communication.Robustness. This illustrates the anti-attack ability of the information hiding method. It means that the carrier receives many unintentional or intentional interferences during transmission after the message is hidden, but it can still extract the secret information on the principle of guaranteeing a lower bit error rate to ensure the integrity and reliability of the original message [20]. It can also be called self-healing or error-correcting. Generally speaking, the robustness of the algorithm and the hidden capacity are mutually restrictive. In other words, the better the hidden capacity, the greater the possibility of the embedded information being destroyed, and the robustness will decrease accordingly.


The commonly used evaluation indicators of steganalysis include accuracy, applicability, and complexity [21,22].


Accuracy. This is the most important quantitative evaluation index of the steganalysis methods, and directly reflects the ability of the steganalysis methods to distinguish the carrier, including accuracy, false-positive rate, true-positive rate, and so on.Applicability. This is also known as universality and scope of application. Different steganalysis methods are suitable for distinctive steganography methods. Generally speaking, if a method can detect more steganographic methods and apply them to more types of encoders, it is said that its applicability is better.Complexity. When designing steganalysis methods, it is necessary to consider factors such as software and hardware costs, calculation costs, and time costs. Generally, the lower the complexity of the algorithm, the easier it is to obtain various implementable resources for the algorithm, which also means higher practicality.


For the performance analysis in the table, we use “√” to indicate that the performance of this method in a certain aspect has been improved compared with the previous method. Because of the diversity of evaluation coefficients in the evaluation indicators, there is no specific standard to measure the performance of all methods. We can only perform comparisons in a one-dimensional environment, such as FCB or Internet Protocol. When evaluating whether the performance has improved, we only need to compare one or a few of them.

## 3. Steganography Based on VoIP

After the discussion in the first two sections, it can be seen that as a new steganographic carrier, VoIP has the advantages of multiple steganographic regions and difficulty of detection. In line with the different embedding areas of mystery messages, the steganography approaches based on VoIP can be classified into two kinds: voice payload steganography and protocol steganography. Each category can be divided into three subcategories. In this section, we will summarize the existing steganography algorithms and make a clear comparison based on the three indicators: imperceptibility, hidden capacity, and robustness. Imperceptibility can be described in terms of time domain, frequency domain, speech spectrum, speech quality, etc. The evaluation coefficients can be PESQ (perceptual evaluation of speech quality), SNR (signal-to-noise ratio), etc. Hidden capacity can be determined by BPF (bits per frame), BPS (bits per second), etc. Robustness can be evaluated by various coefficients, such as TER (test error rate) and ADR (accurate detection rate). The evaluation coefficients involved in the steganography algorithms are shown in Table 3.

### 3.1. Steganography Based on Voice Payload

The steganography methods based on voice payload have better imperceptibility and larger hidden capacity [18,23]. At present, many steganography algorithms are based on the voice load part. The mainstream approach is to make use of the redundancy of the voice stream itself and complete the covert communication by embedding secret information in the redundant bits of the carrier voice stream. Judging from the existing literature, the main steganography methods based on the speech payload are as follows: steganography based on fixed codebook, linear prediction coefficient, and adaptive codebook.

The VoIP steganography distribution diagram based on the voice payload is presented in Figure 4 [24]. This figure describes the coding procedure of speech. First, the original speech signal is preprocessed, and the linear prediction coefficients obtained are converted into line spectrum pair (LSP) parameters and quantized. The quantized LSP forms a synthesis filter. The adaptive code vector and the fixed code vector are respectively taken from the ACB and the FCB, multiplied by the gains.

The sum of G_a_ and G_b_ is taken as the excitation signal and input to the synthesis filter. The steganography algorithm based on voice payload is carried out in various steps.

#### 3.1.1. Steganography Based on FCB

The fixed codebook characterizes the excitation of aperiodic components in the speech signal, and this is an important part of the speech encoder. It occupies a relatively high proportion of each speech frame, and its structure is based on the interleaved monopulse arrangement design and uses a non-exhaustive depth-first tree [25] algorithm in order to search for the optimal solution, so there is a large redundancy space and more opportunities to embed the secret information [26]. Table 4 summarizes the steganography methods based on fixed codebook.

Tahilramani et al. [27], in 2015, proposed the concept of secret information steganography within the structure of an exciting codebook using ACELP [1]. First, the temporary code vector is obtained by selecting the pulse position with the greatest amplitude. Then, the codebook vector replaces the pulse of every track. If the value of the codebook vector increases after the replacement, a new code vector is obtained. If not, the previous code vector is stored. A non-zero unit pulse is assigned as the pivot pulse in each track, and the secret data is embedded in the negative pulse position code. The fulcrum pulse position technology can also be used to allocate a smaller number of bits besides hiding information in the excitation code vector coding, thereby reducing the decoding requirements of the code excitation vector at the decoder and enhancing the concealment.

Tian et al. [28], in 2016, presented a codebook segmentation method based on Neighbor Segmentation (NID). First, the fixed codebook is categorized, and a key is added to the preprocessing; then, the mystery message bits are converted into binary k-ary. After the information has been buffered, 3k-ary digits are formed and embedded in the grouping codebook to form a quantization. The index sequence, through post-processing, becomes a voice stream with embedded secret information. Unlike the complementary neighbor vertex (CNV) algorithm, the CNV quantified the codebook into points to ensure that neighboring points are in the opposite state, and NID divides the codewords of the adjacent index into independent sub-codebooks (partitions), which is simpler, safer, and increases the embedding capacity. Moreover, a flexible multi-value segmentation method is introduced, which is more appropriate for the practical application of covert communication.

Yan et al. [29], in 2016, introduced the fixed codebook search procedure of the G.729 codec and proposed that diverse pulses have distinctive positions in the fixed codebook vector, and the position correlation between contiguous pulses can be recycled and the parity of the position value in the fourth pulse embeds secret information. In the fixed codebook search process of every subframe, four pulses are selected from the fixed codebook vector, and the pulse position and pulse flag are selected according to the pulse. The encoding position of adjacent pulses is transposed to realize the embedding of secret information. This method has better imperceptibility, real-time operation, and security, but this method is only applicable to the G.729 codec, which limits the universality of the algorithm.

Ren et al. [25], in 2018, put forward a steganography algorithm based on AMR (Adaptive Multi-Rate) fixed codebook search standard and non-zero pulse position correlation [30]. Firstly, calculate the optimal probability of pulse, pulse correlation, and hit function value in the fixed codebook search. The embedding cost could be obtained on the basis of the former two, and then the additional distortion could be calculation on the basis of hit function value and the embedding cost. The optimal fixed codebook vector would be output after selecting the minimum additive distortion in line with the preprocessed secret information and syndrome-trellis codes. Experimental results show that, compared with other steganography algorithms [31], this steganography scheme not only has better hearing concealment and safety, but it also has a very large hidden space.

Ren et al. [32], in 2019, put forward a new reliable steganography method using the characteristics of pulse distribution. Firstly, the preprocessing operation is carried out when the secrecy message is embedded, while the embedded message is also divided into two parts: the original secret information, and the marked information. Then, the embedding rule is determined according to the principle of the smallest change in the pulse distribution characteristics, which guarantees that the possibility of the pulse locus on a similar path remains unchanged. In addition, the distribution is more unplanned, such that the steganographic audio is closer to the original audio in the pulse distribution. This steganography method enhances concealment and anti-steganalysis ability by designing the embedding process and random mask. However, the steganographic capacity requires further enhancement.

#### 3.1.2. Steganography Based on LPC

Due to the correlation between voice samples, a voice sample can be approximated on the basis of the last few voice samples, or their linear combination. A distinctive set of prediction coefficients is determined when the error between the actual voice and linear prediction samples reaches its minimum under the minimum mean square error criterion. The linear prediction coefficients must be converted into LSF (Line Spectrum Pair) coefficients before encoding, and the LSF coefficients of each frame must be quantized using the split vector quantization (SVQ) method, which provides space for the hiding of confidential information. The most commonly used steganography algorithm based on linear prediction coefficients is quantization-index-modulation (QIM). The steganography methods based on LPC are generalized in Table 5.

Liu et al. [33], in 2014, proposed a joint matrix coding and linear prediction speech coding hiding method based on QIM. First, the speech code stream is regarded as an LPC filter sequence, and matrix A represents all possible embedding positions when using QIM for embedding. Then, matrix A is divided into blocks, and n frames are extracted from each block area in combination with chaos theory for embedding, and matrix B, which is to be embedded, is obtained. The smallest embedding unit is obtained by dividing B into blocks, and the secret key is used to select the embedding position [26], and a sequence of positions to be embedded will be obtained, in combination with chaos theory. The experimental outcomes of this scheme indicate that [34] this method has the advantages of the lowest distortion and high concealment under the same embedding capacity, when compared with existing methods.

Addressing the problems of unsatisfactory steganography performance in [35,36] and large degradation of call sound quality, Huang et al. [37], in 2015, proposed a HOOK mechanism steganography model. The essence of HOOK is that it can be applied through system calls. The difference between it the previous two methods is that this steganography scheme attaches two “hooks” to the sending of the communication software. The first “hook” is used to intercept the voice stream before encoding, thus obtaining the original PCM data. The second “hook” hooks up at the sender compress and encodes the carrier data, as well as the secret information to be embedded. In addition, the original RTP data packet is replaced with the encoded information. Finally, the interactive hidden communication process is achieved after sending the modified data packet to the receiver. The steganography model is suitable for the current covert communication of instant messaging software, and has good steganography capacity and rate, which expands the application range of information steganography based on VoIP.

Li et al. [38], in 2017, proposed a secret information embedding method in the linear predictive coding procedure based on matrix embedding. First, the mapping table is built on the basis of the minimum distance of the linear prediction coefficient vector before and after the mystery message is embedded; then, the embedding position and the cover frame are chosen, along with the privileged key and the template. The original codeword is obtained by partially encoding the original voice data of the selected frame, and then select the codeword to be modified and embedded is obtained using the key bits and the Matrix Embedding (ME) technique. The unselected codeword directly enters the encoding process, and the chosen codeword will be altered to its best replacement codeword as per the mapping table. The performance of this method was evaluated on the basis of two aspects: the speech quality distortion after the secret information was embedded; and the security of steganalysis. In addition, the experimental outcomes displayed that this technique had a lower voice distortion rate and a higher degree of security [39].

Yue Peng [40], in 2017, determined the cepstrum distortion cost function of linear predictive coding based on the universal wavelet relative distortion function. Firstly, the LPC coefficient of the speech, as well as its cepstrum coefficient, were determined. Then, the LPC cepstrum single-point distortion cost was obtained, and the coefficient modification position of the speech frame was determined, as well as the distortion cost function and Syndrome-Trellis Codes (STC). The secret information is embedded in the position at which the distortion cost function is small. Compared with the direct matrix embedded steganography scheme and the simple LSB replacement steganography scheme, the imperceptibility of this method is significantly improved.

Liu et al. [41], in 2017, put forward a QIM steganography technique based on the replacement of quantized index sets in linear predictive coding. This technique regards every quantized index set as a spot in the quantization universe and performs steganography in this field. First, the points in the quantization index space are divided into eight groups on the basis of the genetic algorithm. Every cluster symbolizes a three-digit binary amount. In addition, the initial voice frame is partially coded to get the initial index spot and clarify which cluster the spot belongs to. If the initial index spot is in the collection of the mystery bit, no substitute is required; otherwise, the original index point has to be substituted with the closest replacement point in the related secret bit collection, and the replacement point and the original point have only one quantization index different. In this way, when three binary bits are hidden, at most one quantization index needs to be altered. Compared with former approaches, this method presents an express improvement with respect to embedding efficiency and detection resistance.

The traditional QIM-based steganography algorithm divides the codebook into two parts, and searches for codewords in different codebook collections on the basis of whether the embedded mystery message is “0” or “1”. Based on the consideration of the optimization of the codebook division, Huang et al. [42], in 2017, put forward a novel QIM control steganography algorithm. The essential design of this technique was aimed at establishing a graph model of the codebook area of the quantizer to ensure that each codeword and its neighboring codewords were in the opposite state, and a distortion boundary was set. Performance testing and steganography experiments showed that the proposed steganography scheme was safer and stronger than the traditional QIM technique and the original codebook technique.

Anguraj S et al. [43] put forward a steganography approach based on an optimized audio embedding technique (OAET) in 2019. First, the original voice information and secret information were converted into a binary data stream. Then, the OAET algorithm was used to embed the clandestine message Mbit into the original voice information, compare the original voice stream bit stream with the Mbit, and change the left and right bit values of the Mbit in the original voice stream in response to different scenarios. In this way, a voice stream with embedded secret information was obtained. Experimental data showed that, compared with the previous LSB steganography method, this method improved security and imperceptibility. However, this algorithm can only be applied to voices in “.wav” format, and therefore the applicability of this method needs to be further improved.

In 2020, Li [44] proposed an adaptive G.729 voice steganography according to bit-grading. Firstly, the K-means clustering algorithm was used to assess the steganographic property of the bits, with the bits with better steganographic performance being screened out; then, the concealable bits were further clustered to obtain the classification results. Next, to adaptively embed secret information, encoding matrices with different lengths and high embedding rates were selected in accordance with different levels of concealable bits, while the receiving end extracts confidential message in line with the corresponding check matrix. This method maximizes the hiding capacity, while ensuring concealment.

#### 3.1.3. Steganography Based on ACB

The adaptive codebook search is an important part of the speech coding process. Its purpose is to predict the pitch period, and the obtained pitch delay is the prediction result of the pitch cycle [45]. The pitch period is hard to predict precisely, due to the influence of various factors, and the jitter amplitude in the unvoiced segment is high. The randomness is obvious, and it has a large redundancy. Therefore, many steganography methods embed mystery information by regulating the pitch cycle, so that the resulting detriment to voice quality is small, and the concealment is good. Table 6 sums up the steganography algorithms based on the adaptive codebook.

Liu et al. [46], in 2013, addressed the problem of low-rate speech coding and proposed an information hiding technique using pitch prediction [47]. In the pitch prediction coding process, the embedded mystery message is realized by controlling the scope of adaptive codebook search. First, the clandestine information is converted into a confidential information bit stream through preprocessing, and is then embedded in the voice frame after being encrypted, so that the hiding of the information is achieved while voice compression is performed. The algorithm has good concealment and low computational complexity.

Yan et al. [48], in 2015, put forward a double-layer steganography algorithm. Their pitch cycle search set contained four consecutive elements. Using the feature of parity between adjacent integers, the first layer of steganography can be realized by adjusting the pitch cycle of the first and third subframes. According to the arbitrariness values of the modified pitch period in their respective sets, the second level of steganography was achieved by searching for the optimal pitch cycle combination. In the double-layer embedding process, the embedding secret information could be realized by simply adjusting the value set of the pitch period; the embedding process and the speech coding process were closely integrated. The algorithm divided the embedding process into two layers for processing with a small modification range, thus improving the hiding capacity, and offered superior real-time performance. It was able to withstand the discovery of the steganalysis algorithm [26], but did not greatly improve voice quality.

Regarding the security of [49] and insufficient output voice quality, Artur Janicki [50], in 2016, proposed a better form of the IP phone steganography process, called HideF0. Firstly, the mean square error is obtained from the real pitch parameters and approximate parameters. If the error is bigger than the set line, the original pitch parameters are output directly; if the error is less than the set threshold, then the first three approximations are deleted and replaced with secret information to achieve the purpose of hiding communication. This method uses the voice data packet header, and the quality and safety of the output voice are significantly improved.

Yang [51], in 2017, considering the unvoiced and voiced pitch delay features of AMR speech, found that the pitch delay of the voiceless section has no short period relative stability and large redundancy, and proposed an adaptive steganography algorithm based on the unvoiced pitch delay jitter characteristics. The algorithm adaptively selects the unvoiced sub-frames by using the pitch difference distribution of consecutive odd-numbered sub-frames and performs steganography of the mystery message in the voiceless section in line with the steganography guidelines, in order to avoid damage to the short period relative steadiness of the pitch delay of the voiced section. The experimental outcomes indicated that the method had excellent auditory concealment and statistical security.

Liu et al. [19], in 2019, proposed a novel steganography plan on the basis of decimal pitch delay search. To obtain better steganographic performance, this scheme embedded mystery messages into the decimal pitch delay, while the integer pitch delay parameter remained unaffected. The covert information was encrypted first, and then the partial similarity between the mystery message and the decimal pitch delay was calculated, which determined whether to embed the secret information according to the decision threshold. All decimal pitch delays were used as substitutable coverage bits to attain the largest embedding capability. Additionally, APMS [52] adaptive partial matching steganography has also been discussed, improving the security of the algorithm.

Liu [53], in 2020, proposed a scalable matrix steganography method for enhanced speech service coding. This method combines adaptive codebook partitioning and scalable matrix steganography. First, the relative search pitch delay parameter and the decimal pitch delay parameter are extracted from the speech stream and treated as embedded objects. According to the parity, these parameters are divided into two codebooks, representing the embedded information “0” and “1”, respectively. Then, the secret information is divided into blocks, with the allocated bits of each block being required to be equal; then, the Hamming check matrix of each block is calculated, and finally the index value is calculated. If the index value is equal to 0, there is no need to modify the carrier information; otherwise, the index bit of the carrier is modified. Theoretical analysis and experimental results show that, compared with the existing methods, this method not only improves the embedding efficiency, but also has better steganographic transparency and bandwidth. In addition, the method can be applied to 5G and other ACELP-based audio coding environments.

In summary, an increasing number of researchers are focusing on the study of covert communication using VoIP as the transmission carrier, especially steganography methods based on the voice payload. In addition, the existing research results represent a great improvement on the traditional least significant bit replacement method in terms of hidden capacity, imperceptibility, security, and so on. A common point in existing low-rate speech stream steganography methods based on speech stream redundancy is that the redundant bits in the low-rate encoded speech frame are treated equally. However, the impact of each frame bit in the coded speech frame on the quality of reconstructed speech is not equal. Therefore, the issue of how to use the redundant bits of low-rate speech frames to more effectively realize steganography algorithms with perceptual transparency requires further study.

### 3.2. Steganography Based on the Protocol

Network protocols, including the application layer, transport layer, network layer, and link layer, are usually developed at different levels, and each layer is responsible for distinct communication functions. The link layer usually contains the device driver in the operating system and the corresponding network interface card in the computer. They deal with the niceties of the physical interface with the cable in a cooperative fashion. They are usually generated automatically by the system itself, and generally cannot be changed in design, so information steganography cannot be performed at this layer. The network layer handles the activities of packets in the network, while the transport layer chiefly affords end-to-end communication for the applications on the two hosts; the application layer is responsible for handling particular application specifics. This makes it possible to embed secret information.

Information steganography technology that uses the network protocol uses the network protocol header as the carrier to hide confidential information in network data packets for the communication of the mystery message. The principle is to use the undefined, reserved, optional, and other domains in the network data packet and the distinctive time flow, sequence, quantity, arrival time, and other features of the data packet to establish covert communication between different hosts on the network and to transport the secret information. Specifically, this can be classified into three categories [54]: Steganography based on the network layer, the transport layer, and the application layer. The information hiding technology based on the TCP/IP network protocol is based on the redundancy or optional fields in the header of the network protocol and the loose restrictions of network equipment [54]. Without adding additional bandwidth, it is difficult to detect for network firewalls and interruption detection structures, and it can easily evade network monitoring to achieve the purpose of information hiding. The network protocol includes a link layer, a network layer, a transport layer, and an application layer. However, the communication protocol of the link layer is normally generated automatically within the system, and generally cannot be changed or designed. Therefore, the research and discussion of the information hiding technology using the TCP/IP network protocol are usually focused on the network layer, the transport layer, and the application layer [26].

The model for the hidden transmission of VoIP information based on network protocol is presented in Figure 5 [15].

When protocol steganography is used for covert communication, the sender embeds secret information in the protocol data packet using steganography algorithms (Steg) to obtain the secret data packet. The secret data packet can be transmitted through various protocol layers. The receiver can extract the secret information using steganalysis algorithms (Extr). On the basis of the model depicted above, this can be approximately divided into four scenarios (assuming that the extraction process is successfully able to extract the secret information), as shown in Figure 6 [16].

These four scenarios are all end-to-end communications. First of all, scenario A is similar to scenario D, with the secret information embedding and steganographic data packet being performed at the sender. The cover communication and steganographic communication are synchronized. Scenario A obtains the mystery information at the receiving end, but scenario D extracts the mystery message during the communication process, and the receiving end obtains the common data packet or the damaged data packet directly. In scenario B and scenario C, the cover communication is performed first, and then the common packet is embedded with secret information to form a steganography packet for transmission. The second scenario culls the secret information at the receiving end, and the third scenario culls the confidential information during the communication process, which is similar to scenario D.

The extraction process can occur at any time after the confidential information is embedded.

#### 3.2.1. Steganography Based on the Network Layer

The mechanism of information hiding algorithms in steganography based on using the network layer protocol as the carrier involves placing confidential information that needs to be hidden in areas that network monitoring and detection either ignore, of experience difficulties in detection. Nowadays, many steganographic algorithms use the IP identification field as a carrier to hide information to achieve covert communication. At the sending end, the sender converts the secret information that needs to be sent into ASCII code, then encrypts it, and converts the encrypted information into the value of a seemingly legitimate IP identification field according to the corresponding algorithm in order to deceive IDS (intrusion-detection system), firewalls and other network security equipment, thus achieving the use of IP identification domain fields to hide secret information and realize covert communication [55].

Steganography algorithms based on the protocol are presented in Table 7.

Huang and Tang [56], in 2016, proposed a new steganography model of covert communication space based on network voice. Based on the space model, a quick-start retransmission mechanism technique was designed to solve the packet loss problem. The time and space negotiation mechanism makes it possible for communication parties to share the current used hidden vector through a secret channel. Only some media packets in the media stream of the sender are used to hide data, and the receiver needs to identify the streaming media that is carrying secret information, and then determine which hidden vector is being used to embed the secret information [57]. When the receiver knows that the hidden vector used for embedding the secret information is in the streaming media packet, the secret information can be extracted directly. This method solves the problem of improving imperceptibility, hiding capacity, and synchronization efficiency without affecting channel concealment.

#### 3.2.2. Steganography Based on the Transport Layer

Steganographic information hiding techniques based on the transport layer protocol are mainly based on transmission control protocol (TCP), user datagram protocol (UDP), and real-time transport protocol (RTP) in order to realize information hiding [58]. The function of TCP is to ensure that all packets delivered to the destination application are in order, without packet loss or errors. UDP sends individual data from the application to the IP and routes it to the remote end. RTP mainly provides network transmission services for real-time applications.

Gong, in 2015 [59], proposed an information hiding method based on IP phone transcoding by compressing public information in order to save space for information hiding. First, the payload of the RTP protocol header is analyzed, and then it is decided whether to detect the user’s voice that is carried in the RTP packet and whether to encode the original voice. Then, an appropriate codec is selected for public encoding, and a voice stream is generated that has similar quality to the original voice, but with a smaller payload than the original voice stream. Finally, the voice stream is transcoded into the original payload field, and the remaining space can be used for hiding information. The results proved that this technique improves hiding capacity when compared with previous hiding technologies based on VoIP, and it is hard to detect.

Jiang et al. [60], in 2016, proposed a UDP-based VoIP communication scheme. First, a prediction model is established based on fractal interpolation to determine whether the VoIP packet is suitable for data hiding. If it is unsuitable, the original data of the data packet will be retained. Otherwise, the data embedding algorithm of the variable embedding interval of the advanced encryption standard will be hidden first, and secret data will be encrypted using a block cipher. Later, the data are divided into multiple groups, and each group is embedded in the VoIP stream data packet. Then, the Gilbert model is used to simulate the actual network environment to deal with the loss of data packets. The experimental data indicate that as the degree of packet loss increases, the mean-variance of the voice quality metric (PESQ score) between the “unembedded” voice samples and the “embedded” voice samples gradually decreases, and the security of secret data is also improved.

X. Lu et al. [61], in 2016, put forward a network steganography program based on the length of UDP packets by analyzing the flow of UDP packets and several storage characteristics of data files. First, the sender sends some data packets. The secret information is sent with the length of the data packet due to the randomness of the packet length distribution [26], and then multiple IP addresses are sent through the router. To enhance the security of secret information transmission, some fake packets are added to confuse the monitor. Random coding technology is used for this process, which is better able to simulate usual traffic, thus overcoming the deficiencies of present solutions. Comprehensive experimental outcomes show that [62] the proposed hidden channel is well able to simulate the statistical features of normal traffic and has greater security than existing algorithms.

The StegVAD algorithm proposed by Sabine S. Schmidt and Wojciech Mazurczyk [63] in 2017 improves the channel capacity without affecting the quality of VoIP sessions. This method converts the Voice Activity Detection (VAD)-activated VoIP stream of voice activity detection into a non-VAD VoIP stream. The fake RTP packets are generated by appropriately increasing the sequence number and timestamp during the silent period produced by the encoder. As the carrier of secret embedded information, the monitor is then confused in order to achieve covert communication. Although the channel capacity is improved, the robustness and anti-detection performance of the algorithm is not satisfactory, which is also the focus of future work.

S. Deepikaa and R. Saravanan [64] proposed a hash-based steganography method in 2020. First, the voice stream is obtained from the UDP protocol, and a hash array is constructed from the frame data. For each new frame, the hash array must be updated. Then, the secret information is cut, and the appropriate bit position is selected according to the hash function in order to embed the clandestine message. When the secret message is fully embedded, the hash array value is set to 0. The hash array and audio samples are sent to the receiver as a VoIP frame. The receiver can then extract the secret message based on the hash array flag value. The experimental results indicated that the algorithm offers good performance in the areas of computational complexity, undetectability, and voice quality for the sender and receiver. However, the hash array takes up extra bandwidth in the VoIP communication process.

#### 3.2.3. Steganography Based on the Application Layer

Steganographic information hiding algorithms based on the [58] application layer application layer protocol achieve information hiding mainly because existing firewalls and routers generally do not check the application layer protocol. The uppermost layer of the network protocol model is the application layer. The role of the application layer is to be responsible for the data exchange between the user and the transport layer. Since existing firewalls and routers generally do not check the application layer protocol, this provides a way for the application layer protocol to be used as a carrier for information hiding in order to achieve covert communication [26].

Li et al. [65], in 2013, proposed two information hiding methods based on command exchange and command control on the basis of research into the FTP protocol. The hiding algorithm, which is based on the command exchange, sends instructions stating whether the secret information sent is “0” or “1”, and the receiver assesses the secret information on the basis of the received instructions; The recipient extracts the mystery message on the basis of the order of the instructions [66]. The proposed method possesses good concealment and robustness.

Yao et al. [67], in 2016, proposed a method for hiding information in the FTP protocol on the basis of the study of the file transfer protocol (FTP). First, the sender sends an abort (ABOR) command to indicate the start of communication, and then divides the secret information. Each time, N bits are taken and compared with the coding table to find the corresponding directory name, and sends a change working directory (CWD) command for the directory. Then, a CDUP (change to parent directory) command is sent to return to the higher directory in order to facilitate the next search, and finally an ABOR command is sent to indicate the end of the communication. The experimental results show that [12] the concealment ability of a single command is greatly increased through appropriate coding. When sending a small amount of secret information, the level of concealment is very high, but when a large amount of data is sent, it can be easily detected by statistical software, thus reducing the level of concealment. Therefore, the focus of future research is on improving concealment when transmitting large amounts of secret data.

From the above, it can be seen that there are some steganography methods based on the network protocols that can achieve information hiding using VoIP. However, their comprehensive performance, including the embedding capacity and security of secret information, is not as good as steganography algorithms based on the voice payload. Therefore, it is still necessary to strengthen the research and innovation of steganography algorithms based on the voice payload. This is the focus of future research.

## 4. Steganalysis Based on VoIP

Chapter 3 presented a detailed introduction to steganography methods based on VoIP, which can mainly be divided into two categories: voice payload-based methods and protocol-based methods. As a countermeasure against steganography, steganalysis has been drawing increasing attention. The purpose of this technology is to detect the existence of confidential information, disclose the flaws of current steganography, and estimate the security of steganography. Chapter 4 will summarize steganalysis methods based on the voice payload and the protocol, while also evaluating various steganalysis methods with respect to three different indicators: accuracy, applicability, and complexity. With respect to accuracy, there are many parameters to measure. For example, ACC (accuracy), FPR (False Positive Rate), FNR (False Negative Rate), etc. Indicators such as AC (applicable codec) and ASA (applicable steganographic algorithms) can be used to evaluate applicability. Complexity can be evaluated on the basis of SC (space complexity), TC (time complexity), etc. The evaluation coefficients involved in steganalysis methods are displayed in Table 8.

### 4.1. Steganalysis Based on Voice Payload

Steganography algorithms based on voice payload can be classified according to the parameter domain, which is divided into the fixed codebook parameter domain, the linear prediction coefficient domain, and the adaptive codebook parameter domain. In line with this classification method, steganalysis classification can be divided into steganalysis methods based on fixed codebook, linear prediction coefficient, and adaptive codebook. A distribution diagram for VoIP steganalysis based on voice payload is shown below [24]. Figure 7 mainly describes the speech decoding process. First, the binary code stream is processed for error correction, and the index and gain of the ACB and FCB are used to search for the corresponding codebook vector in their respective codebooks. After weighting by the gains G_a_ and G_b_, the synthesis filter excitation signal is formed, and after passing through the post filter, the synthesized speech signal is obtained. The coefficients of the synthesis filter are linear prediction coefficients converted from LSP parameters. Steganalysis based on voice payload is carried out in this process.

#### 4.1.1. Steganalysis Based on FCB

A fixed codebook vector can obtained using a depth-first tree in encoding. However, this result is suboptimal, so there are other alternatives to the required codebook vector [68]. Using this feature, the existing steganography methods incorporate the steganography operation into the codebook search in order to embed the information. The detection of this steganography method often distinguishes the original sample from the steganographic sample on the basis of differing characteristics between pulses. Steganalysis methods based on FCB are elaborated in Table 9.

Miao et al. [69], in 2014, proposed two methods for detecting various types of compressed domain steganography (CDBS) in ACELP speech. The first is the Markov method, which divides the fixed codebook index into a list of subsequences; there are N indexes in every subsequence [70]. N represents the number of non-zero-amplitude pulses in every track; then, an N-1order Markov chain model is constructed to analyze it. Finally, whether secret information is embedded in the signal is determined by calculating the Markov transition probability (normal signals are relatively smooth, and signals embedded with secret information are relatively sharp). The other is the entropy method. When secret information is embedded in the signal, the interdependence of the combined pulse will change. Then, usual signals and secret signals are detected by calculating the joint entropy and conditional entropy of the signal (the entropy values of signals embedded with confidential messages are lower than those of normal signals) [71]. The two methods offer a great improvement in detection accuracy.

Ren et al. [72], in 2015, observed that steganography schemes based on a fixed codebook parameter domain increase the probability of the same pulse position in the same trajectory. Based on this phenomenon [73], they proposed the Fast-SPP (same pulse position) feature steganalysis method. First, the SPP values for all the tracks that are not at the first pulse position are selected as the steganalysis feature, and then the average of the SPP values is calculated as the final feature. Supervised machine learning methods are used to train and test the features, and subsequently, classification models are generated. The joint possibility of the same pulse position matrix is used as the eigenvector for steganalysis. The main advantage of this method is that it is not only designed for AMR audio codecs; it is also suitable for numerous audio codecs using the ACELP algorithm. The experiments showed that when the embedding rate reached 30% or more, the accuracy of the algorithm reached more than 90%. However, when the embedding rate was less than 15%, this method was not very reliable.

Liu et al. [74], in 2016, put forward a steganalysis technique based on three characteristics: long-term distribution of pulse position, short-term distribution, and correlation between pulses. First, the three values corresponding to the following three characteristics are calculated: the probability and the Markov transition probability of the pulse position, and the joint probability matrix describing the correlation between pulses. Then, the trained SVM (support vector machines) is used to classify the features and to judge whether any secret information has been embedded. This technique greatly enhances the detection of accurateness under any embedding rate and with arbitrary sample length [1]. Specifically, this scheme is able to effectively detect steganography using only a few potential coverage bits, which is difficult to detect effectively using existing approaches.

Huang et al. [75], in 2017, proposed a hybrid steganalysis scheme. The pulse pair features are extracted after grouping and processing the training samples, and they are trained separately by specific classifiers. Then, the pulse pair feature of the test samples is extracted, and multiple pieces of evidence are obtained on the basis of multiple classifiers. Dempster-Shafe Theory (DST) is used to combine of the evidence from multiple specific classifiers to obtain a comprehensive detection outcome. All steganalysis methods are assessed using chosen characteristic sets based on pulse pair statistical features. This method improves the detection accuracy of secret information.

Steganalysis algorithms based on fixed codebooks destroy the correlation between pulse positions. Chen et al. [76], in 2019, proposed a steganalysis method combining RNN and CNN. First, the pulse position sequence of the speech embedded with secret information is quantized as a matrix and input to the RNN; then, an optimal model is obtained by training the RNN. The output of the RNN is used as the input data of the CNN, and key features are extracted by the CNN. The feature relationship is divided into four types: intra-frame, inter-frame, inter-phoneme, and inter-word. Samples with a long time interval can be analyzed because of the introduction of the RNN. The experimental results demonstrated that this method had a higher detection accuracy for short-duration samples with low embedding rates.

Tian et al. [77] put forward a steganalysis algorithm based on multi-classifier fusion for AMR steganography in 2019. First, the pulse pair characteristics-based features (PPCF) and the pulse-correlation-based features (PCF) are extracted. Then, these two sets of features are placed into two diverse classifier musters in order to obtain two kinds of forecast outcomes; later, the second type of forecast outcome is treated as a particular kind of feature, and this is input into the additional classifier. The third kind of forecast result is obtained from the set. Finally, the three forecast outcomes are combined to obtain the ultimate detection outcome. The experimental data indicated that this technique was able to achieve higher detection accuracy when compared with FCB steganalysis methods based on support vector machine (SVM). However, the training and optimization of multiple classifiers was a time-consuming task.

Sun et al. [78] proposed a new adaptive multi-rate encoder (AMR) steganalysis model in 2020. First, the Markov transition matrix is obtained from the original speech signal based on AMR, and then the first feature statistical characteristics of pulse pairs (SCPP) are extracted. Because this feature reflects local features, a convergence feature that reflects global features is introduced. After the effective fusion of these two features, the result is sent to the extreme gradient boosting (XGBoost) classifier. As a result of the training of the model, it is able to reach an optimal state, and then it can be tested. The experiments showed that this method was able to achieve good performance when detecting AMR-based voice streams. However, this method is only applicable to AMR, so its applicability needs to be further improved.

#### 4.1.2. Steganalysis Based on LPC

LPC is an important part of VoIP; LPC parameters need to be converted into LSF coefficients in the encoding process. The LSF coefficients of each frame must be quantized using the split vector quantization (SVQ) method. After quantization, the correlation characteristics of the codeword will change. Many steganalysis algorithms based on linear prediction are based on this feature for detection. Table 10 generalizes steganalysis methods based on LPC.

Li et al. [79], in 2013, discovered that the QIM steganography technique alters the phoneme distribution features in the compressed speech stream; therefore, he proposed a phoneme vector space model and a phoneme state transition model to quantify the phoneme distribution features. Firstly, in order to obtain a phoneme sequence, a segment of speech is divided into some frames, and then a vector is constructed that represents the quantification of the phoneme sequence by searching the phoneme dictionary. A steganography detector is constructed that can detect QIM steganography algorithms based on the obtained quantized features and SVM. Experiments on typical low-rate speech coding standards G.729 and G.723.1 showed that the performance of this method was far superior to existing detection methods. It not only reduces the decoding time of the compressed speech, it also realizes fast and accurate steganographic detection of QIM steganography.

Li et al. [47], in 2017, studied quantization index modulation (QIM) steganography in low-bitrate coded speech streams. A quantized codewords relationship network model based on the segmentation of vector quantization (VQ) codewords in contiguous speech frames is constructed. Firstly, to extract the quantized codebook, partial decoding is implemented on the detected samples; then, the QCCN model is reduced to make a more solid connection network. After the connection features of the vertices in the clipped correlation network are quantified, the original feature vector of steganalysis is obtained. Principal component analysis (PCA) is used to decrease the dimensionality of the original features. Finally, the SVM classifier is used to classify the characteristics to determine whether they are steganography speech. The experimental data indicated that the QCCN steganalysis technique was able to successfully discover QIM steganography when employed with low-bitrate speech codecs such as G.723.1 and G.729.

Lin et al. [80], in 2018, proposed an effective online steganalysis technique for detecting QIM steganography. The technique can be broken down into two parts: training and detection [34]. First, a codeword correlation model is proposed based on a recurrent neural network to extract relevant features, considering mainly the correlation of continuous frames, intra-frame correlation, etc. Then, the characteristic classification model is used to sort related characters into cover and steganography voice. The steganographic and cover speech streams are labeled according to the codeword correlation, and the data are put into the steganalysis model (RNN-SM) for training with a supervised learning framework. After training on large amounts of data, the input voice data is detected to determine whether it contains secret information. The experiments showed that RNN-SM had a high detection accuracy rate on samples with a full embedding rate. Its detection accuracy rate is still above 90% with speech as short as 0.1 s, which is significantly higher than other currently available methods. RNN-SM also achieved higher accurateness on samples with a low embedding rate. The average test time for each sample was less than 0.15% of the sample length.

Han et al. [81], in 2018, proposed the introduction of linear prediction methods in the field of signal coding and speaker recognition in audio steganalysis, resulting in a noteworthy difference between cover and steganography voice. First, the feature extraction function is used to extract features from the data file [26] after the data set is collected, and the supreme value of every character is used as the classification characteristic. Then, 4-fold cross-validation is applied to the training machine, using three dissimilar three-quarter data sets as the training set, and the greatest parameters are found. After the training phase is finished, the best parameters are used to make a decision on the test data (the remaining quarter of the data set), and finally the test results are displayed. The training and choice phases are repeated k times to obtain the average of the choice results. The experimental data indicated that this approach possesses good capability, with an accuracy rate of over 96%.

The symbols in the ABS-LPC low-rate compacted voice code stream have temporal and spatial correlation, and all ABS-LPC low-rate compacted voice steganography methods essentially change the value of the symbol [73]. Therefore, Li et al. [82], in 2019, proposed a general information hiding detection method for multi-class low-rate compressed speech steganography from the perspective of symbols. Firstly, a Bayesian network in units of speech frames containing all symbol information is constructed. The speech frame category is used as the root node for expanding the network, and the correlation index is defined to quantify the strength of the symbol association. Then, the network parameters are determined on the basis of a large number of learning samples, and child nodes are used to obtain the posterior possibility of the parent node [73]. Finally a threshold is set to determine whether it is steganographic speech. This method has a good general steganography detection effect and has excellent performance in terms of time complexity.

In 2020, Wu et al. [83] proposed an analysis method for detecting QIM steganography in G.723.1. First, the distribution and the transmission probability matrix of the original speech signal are calculated, and then these are used as the feature vector. With the aim of improving the performance of the technology, principal component analysis (PCA) is employed to decrease its dimensionality. Then, the covariance matrix is calculated, and the main component is selected on the basis of the contribution rate, that is, the eigenvector. This is input into the support vector machine (SVM), and as a result of training with a large number of samples, this method has higher detection accuracy for QIM steganography. However, for short-duration and low-embedding rate samples, the performance of this method needs to be improved.

To achieve fast and accurate detection of VoIP steganographic voice streams, Yang et al. [84] proposed a fast steganalysis algorithm in 2020. First, the vector quantization codeword is mapped to a semantic space, and then a hidden layer is used to extract the relevant features of the codeword, which is then input into the softmax classifier. The output of the classifier is a possible probability, and then a threshold is set to determine whether the input speech sequence belongs to the category of steganographic speech (ordinary speech). The experimental data showed that even when the speech length was 0.1 s, the average detection time of this method was 0.05% of the sample length, and it could be easily applied for online detection. Because the algorithm pursues simplicity and speed, its accuracy is not very high.

In 2020, Wu et al. [85] designed a steganalysis algorithm based on calibration technology and a hybrid classifier. First, the probability distribution of the quantized index sequence of the pilot spectrum frequency (ISF) is extracted from the speech samples, and then the feature extraction model is used based on the three-layer LSTM network to extract the correlation characteristics of the ISF parameters in the time series. After the above two features are processed by calibration technology, they are better able to characterize the changes of ISF before and after information hiding. Then, this is sent to the support vector machine for training. Finally, the index distribution characteristics of speech and the correlation characteristics of the ISF parameters are sent to the trained model for detection. The experimental data indicated that, when compared with existing algorithms, the method introduced by Wu had obvious superiority at low embedding rates.

#### 4.1.3. Steganalysis Based on ACB

The aim of the adaptive codebook search is to extract the pitch information [19] of the speech and then obtain an optimal adaptive codebook index. The pitch period is a very important parameter of the encoder, and there is a high degree of redundancy in the encoding process. Many steganographic algorithms based on pitch delay (such as QIM) embed mystery information by modifying the original value. It is difficult to detect them because of their higher levels of concealment and bandwidth for concealed communication, and there are relatively few steganalysis algorithms for pitch delay. A summary of steganalysis methods based on ACB is provided in Table 11.

Li et al. [86], in 2014, found that hiding pitch modulation information changes the adaptive codebook’s correlation characters for adjacent speech frames in compressed voice streams. To quantify these correlation characteristics, a codebook correlation network model was designed, and a feature vector sensitive to steganography was obtained based on this model. Finally, based on the obtained feature vector and the SVM, a steganography detector was constructed. The experiments showed that the performance of the method for the typical low-rate voice coding guidelines G.729 and G.723.1 was better than existing detection approaches; furthermore, it was also able to realize fast and effective detection of hidden pitch modulation information [87]. Compared with the complete decoding of compressed speech in [88], the writing algorithm, which only needs to partially decode the speech when extracting feature vectors, achieved better detection results.

Ren et al. [45], in 2017, proposed a steganalysis algorithm for obtaining the second-order difference feature matrix of pitch delay by computing the Markov transition possibility based on the continuity difference between the contiguous pitch delays of the initial speech and steganographic speech. To extract the C-MSDPD feature, the procedure is divided into two paths. First, MSDPD1 is calculated by extracting the pitch delay of the detected voice, and the detected voice is recompressed as a standard voice. Then, its pitch delay is extracted, and MSDPD2 is calculated. Finally, the C-MSDPD feature is calculated. Supervised learning is carried out to train the classifier model, and the model is used to distinguish steganographic and cover speech. The experimental outcomes indicate that the capability of CMSDPD is superior to previous methods, particularly when the embedding rate is less than 30%. Owing to the similarities between AMR encoder and CELP encoder, this algorithm can also be applied with CELP encoders, such as G.729 and G.723.1.

Ren et al. [89], in 2018, found that the existing steganography algorithms by modifying the pitch delay [90] would disrupt the short-period relative stability to a certain extent. The first-order Markov transition possibility feature of the subframe distinction combined with the second-order differential Markov transition probability feature of the pitch delay [91] was proposed, and an AMR steganalysis algorithm for pitch delay correlation was designed based on [86]. The sample is classified into two parts [34]: a training sample and a test sample. The first-order and second-order Markov transition probabilities calculated in the training samples are used for training with SVM, and then the data in the test samples are tested. The experimental data in [89] indicate that the detection property of the algorithm at any embedding rate is better than existing steganalysis algorithms, and the improvement in performance is particularly obvious at low embedding rates.

Huang [92], in 2019, focused on the problems of excessively high feature dimensions and insufficient expression of pitch delay characteristics of AMR speech in the existing research, and put forward steganalysis based on the statistical characteristics of pitch delay based on [86]. This method carefully filters the existing features and proposes low-dimensional but efficient second-order difference statistical features of pitch delay, while also introducing a parity statistical feature to make up for the lack of expression ability of the second-order distinction statistical feature of pitch delay value. The properties of the proposed method are evaluated on the basis of a large number of samples and compared with existing methods. The experimental outcomes indicated that this technique could obtain superior detection results when compared with existing approaches under dissimilar embedding rates and distinctive sample lengths [34,91]. The detection of different steganography methods also achieved a high level of accuracy.

Hu et al. [93] proposed Steganalysis Feature Fusion Network (SFFN) in 2020 for the purpose of exposing steganography methods through the combination of quantization index modulation (QIM) and pitch delay modification. It includes three network structures. The feature learning network digs out important features from the codeword and pitch delay input; the feature fusion network combines the features extracted by the previous network to form representative characteristics for the ultimate forecast; and the category network is used to classify the features to determine whether secret information is embedded in the voice. The experimental data showed that this method had superior performance when detecting QIM and pitch delay modification, and it satisfies the requirements of immediacy. It only takes 0.34 ms to detect 10 ms voice samples.

Tian et al. [94], in 2020, proposed a steganalysis method based on basic frequency statistical features. First, the original zero-crossing count (ZCC) average value is extracted from the silent frame of the original voice signal, then the voice signal is recompressed to obtain calibrated voice samples, and then the calibrated ZCC average value is extracted. The difference between the two is taken as the first feature. Then, the Mel-frequency cepstral coefficients (MFCCs) of the silent frame are computed as the second feature. The feature set of the training sample is sent to the support vector machine (SVM) for training, and then, the feature set of the test sample is put into the trained classification, and whether the input sample is a steganographic voice is predicted according to the output result. This technique is able to accomplish higher detection accuracy at any embedding rate, even if a very short silent frame is input. However, the training of the model requires a great deal of samples and parameter optimization, which will take more time.

### 4.2. Steganalysis Based on Protocol

Network protocol steganography can be classified according to the domain of the steganography layer and can be divided into network layer, transport layer, and application layer steganography methods. In line with this classification method, steganalysis can be classified into three categories: steganalysis based on the network layer, the transport layer, and the application layer. Steganalysis algorithms based on the protocol are compared in Table 12.

#### 4.2.1. Steganalysis Based on the Network Layer

Wang [95], in 2009, introduced information entropy into SVM modeling and proposed an information entropy SVM model for detecting hidden channels of ICMP loads. First, a portion of the samples is randomly selected from among all of the sample sets for training, and an appropriate threshold is selected after calculating the entropy value of each sample; then, the training samples with information entropy of less than a certain threshold are discarded in order to obtain a reduced sample set for training a small-scale vector machine. Finally, the data to be detected are collected and preprocessed, and input into the information entropy SVM model after the data have been normalized. The experimental results indicated that the use of the information entropy SVM to detect ICMP load hidden channels had a faster classification speed and a higher classification accuracy, thus also greatly reducing the training time and solving the problem whereby the standard SVM cannot handle large-scale training sets well.

#### 4.2.2. Steganalysis Based on the Transport Layer

Zhao and Shi [96], in 2013, analyzed the hidden information in the TCP/IP protocol and proposed a novel technique for detecting the presence of covert information in TCP initial sequence numbers (ISNs). First, the unidimensional ISN input sequence is extracted from the data packet, and then the phase space reconstruction technique is used to convert the one-dimensional asymmetric sequence into a set of four-dimensional vectors to construct the feature matrix. Then, the second- and third-order statistical features are calculated. Finally, a trained SVM classifier is used to classify its features in order to detect whether the input information is normal or steganographic. The simulation data indicated that the proposed detection technique was superior to existing technologies with respect to detection accuracy, and greatly reduced computational complexity.

Artur Janicki and Wojciech Mazurczyk [97], in 2014, proposed a steganalysis technique based on the Gaussian mixture model and Mel-frequency cepstral coefficients (MFCC) for transcoding steganography detection, and testing different explicit/recessive codec pairs in the double-transcoding single-code supervisor scenario. First, the MFCC coefficients of the received speech signal that are able to describe the frequency spectrum characteristics of the speech well are extracted; then, the Gaussian mixture model is employed to calculate the GMM scores of normal speech and steganography speech, and ultimately detect the latter. The proposed method allows the effective detection of some codecs (such as G.711/G.729), while some other encoders are still more robust to detection (for example, AMR).

#### 4.2.3. Steganalysis Based on the Application Layer

With the widespread application of session initiation protocol (SIP), hiding confidential data in certain SIP header fields has become a potential threat in many applications. Zhao and Zhang [98], in 2012, applied chaos theory to dissect conventional SIP traffic and proposed a characteristic model for detecting hidden data in SIP header fields. First, SIP tags are collected through call termination, and the delayed coordinate method is used to reconstruct the phase space to construct a feature model. In the steganography process, SIP tags are used as the carrier of secret information, so the detection end first calculates the three-dimensional vector of each tag, and later obtains the distance vector between the vectors in the reconstruction space. Then, a comparison is made to determine whether it contains steganographic information after calculating the third-order feature value and the threshold [99]. The experimental results showed that the computational complexity was low, and was appropriate for online operation. However, this method is only suitable for the detection of the steganographic domain of SIP tags, so the applicability needs to be further improved.

Xu et al. [100] put forward a detection method based on multi-feature classification in 2019. First of all, to increase the difference between data streams and to reduce the impact of jump data, it is necessary to preprocess the message data stream. Then, each window is divided into w pieces of information. After embedding the secret information, it is divided according to the same method, so that the normal and secret data for classification and recognition are obtained. Then, the mean, variance, and histogram features of normal communication and hidden communication are extracted, and AdaBoost is used to train and detect the feature data. The outcomes indicated that when the observation window of the introduced technique was greater than 1000, the recognition accuracy reached 96%. However, this method is only suitable for steganography of messages based on the BitTorrent protocol.

Xing et al. [101] designed a detection algorithm based on multi-feature classification in 2020 on the basis of the network steganography technique of the Piece message. First, the normal and steganographic data streams are extracted, the two sets of data are mixed into the data set to be tested, and they are divided according to the window size w. The entropy, mean, and variance of the inter-packet delay in each window are extracted. The identifiers “1” and “0” are used to mark normal data and encrypted data. The data to be tested is divided into several windows of size w, and the data to be tested is classified by the SVM classifier. The experimental data showed that the average value of the accurate detection rate of this method reached 94.7%, but the applicability still needs improvement.

## 5. Future Work and Challenges

VoIP systems, as a service of IP networks, minimize communication cost to the utmost extent, while also meeting people’s various needs such as voice and fax. In addition, to better serve people, more added services are being developed based on it. These advantages have led VoIP to become the mainstream of streaming media communication. However, the security problems of VoIP have always been worrying, and this is also the reason why VoIP applications are not ideal. To advance the security of communication, information steganography based on VoIP has become the focus of research. However, if steganography technology is used by criminals, such as the use of steganography to attack or leak privacy, information security will be threatened. The research on steganalysis is even more important. For VoIP-based steganography, there are the following two main aspects.


Ensuring communication quality. Hidden capacity is very important in steganography, but increasing the hidden capacity often reduces the concealment of communication. Many researchers have successively proposed a lot of algorithms for improving steganography performance, including multi-layer steganography to improve bandwidth use and expand the embedding capacity of secret information. To improve the efficiency of covert communication, more steganographic algorithms with superior performance need to be developed.Improving the detection resistance of the steganography algorithms. Detection resistance is an important index for evaluating a steganographic algorithm and represents the security performance of the algorithm. The purpose of steganography is to hide the possibility of secret information transmission. However, many current steganography algorithms purely pursue the improvement of embedding capacity, while ignoring the security of communication, resulting in many steganography algorithms being tested during the experiment. Therefore, research into detection resistance in steganography algorithms remains a problem that should not be underestimated.


For the steganalysis of VoIP, the problems faced mainly include the following two aspects.


Enhancing the accuracy of steganalysis algorithms. All current steganalysis algorithms aim at high accuracy. However, it is still a big challenge to quickly detect whether steganographic information is contained at low embedding rates and with short sample times [17]. More and more steganography algorithms are beginning to optimize the classifiers to extract more reliable features and try to use the neural network. Although the accuracy has been significantly improved, there is still a lot of room for development.Elevating the universality of steganalysis. Universality not only refers to detecting particular types of steganography method, but also to detecting the steganography of different encoders. At present, most steganography algorithms are aimed at the detection of a certain type of steganography algorithm (such as QIM) or a certain type of encoder (such as AMR) for steganography detection. There are also some comprehensive detection methods, such as steganalysis algorithms based on IP, ICMP, and other multi-protocol steganography techniques. However as far as the field of secure steganalysis is concerned, this is far from sufficient. It is necessary for more researchers to conduct in-depth studies to improve the universality of steganalysis.


## 6. Conclusions

With the increasingly prominent information security issues of VoIP, research into steganography and steganalysis based on VoIP has become a heated research topic. So far, a great number of representative research results have been published. However, the current research based on VoIP steganography and steganalysis is generally not yet mature, and the complete theoretical system and technical framework need to be improved. This article first introduces the concept of steganography, steganalysis, and VoIP, while also reviewing the research on steganography and steganalysis based on VoIP in recent years, dividing it into two categories on the basis of the embedded information area (steganography and steganalysis based on protocol or voice payload) for the purposes of discussion, and then subdividing them into steganography and steganalysis based on FCB, LPC, ACB and network layer, transport layer, and application layer, in order to evaluate and summarize the existing results based on six indicators (imperceptibility, hidden capacity, robustness, accuracy, applicability, complexity), and then analyze the problems to be solved and the challenges based on it. Finally, a summary and outlook with respect to future work and development trends in the field were provided, considering the present study status.

## Figures and Tables

**Figure 1 sensors-21-01032-f001:**
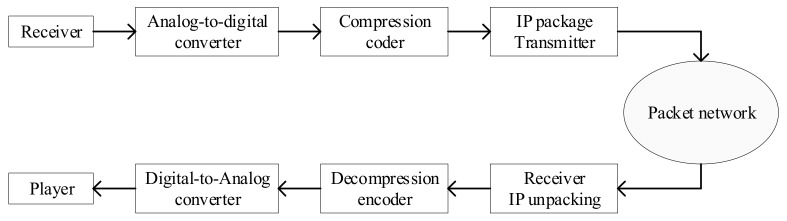
The basic process of VoIP transmission.

**Figure 2 sensors-21-01032-f002:**
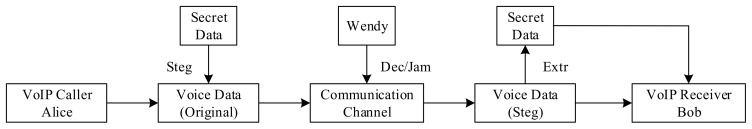
Steganography and confrontation model of VoIP communication.

**Figure 3 sensors-21-01032-f003:**
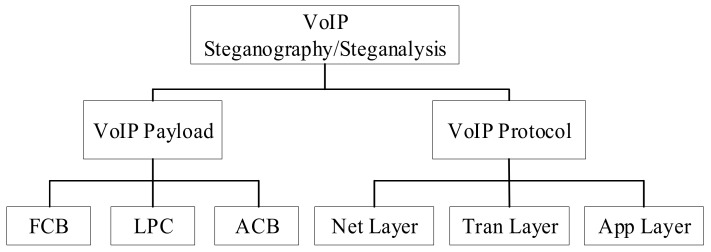
Classification of steganography and steganalysis based on VoIP.

**Figure 4 sensors-21-01032-f004:**
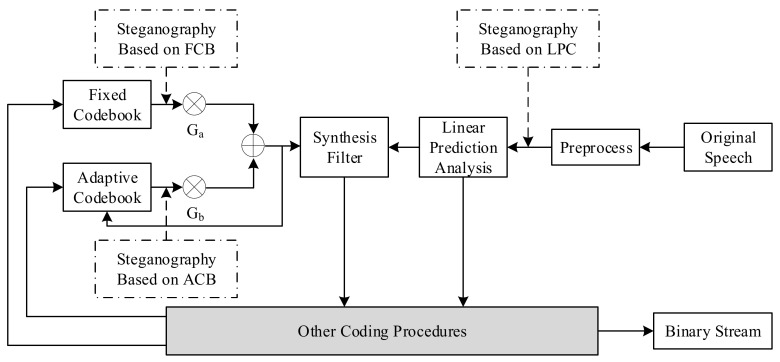
VoIP steganography distribution map based on voice payload.

**Figure 5 sensors-21-01032-f005:**
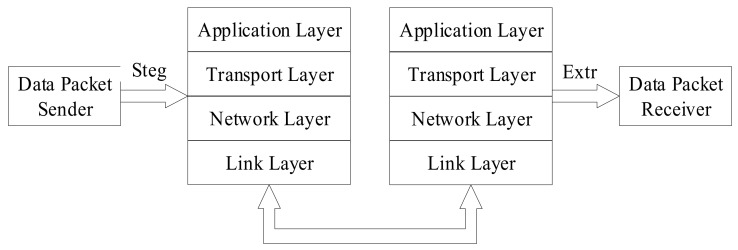
Model of the transmission of hidden information VoIP based on network protocol.

**Figure 6 sensors-21-01032-f006:**
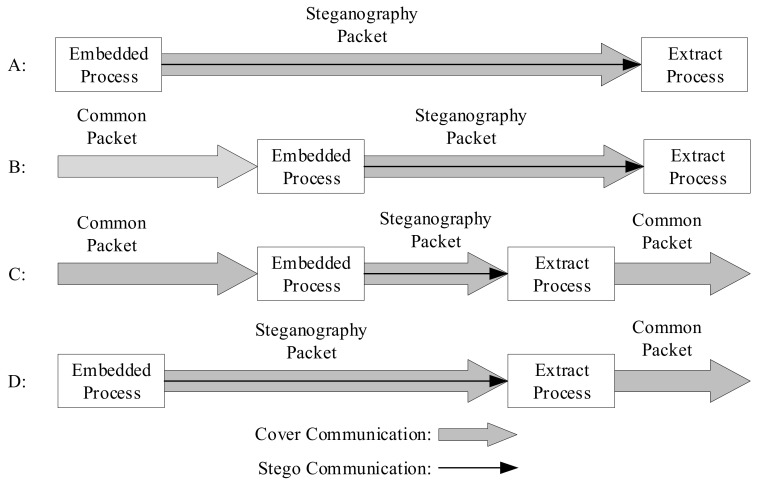
Covert communication scene model.

**Figure 7 sensors-21-01032-f007:**
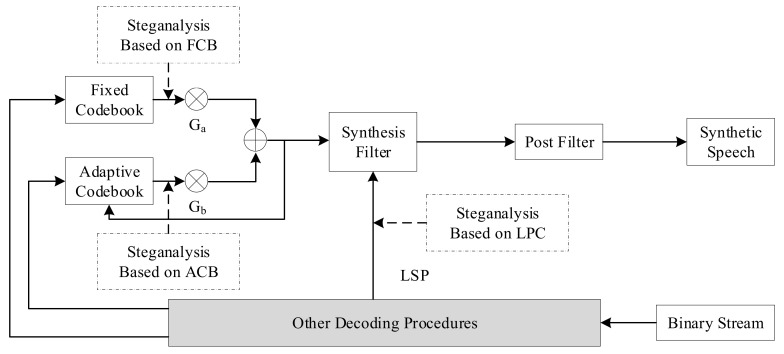
Distribution map of VoIP steganalysis based on voice payload.

**Table 1 sensors-21-01032-t001:** Comparison of steganography and encryption.

Technique	Goal	Security Principle	Attack Type	Technical Method	Application Field
steganography	sc ^1^	cac ^2^	steganalysis	std ^3^ and m-b ^4^	avi ^5^
encryption	dp ^6^	dian ^7^	cryptanalysis	trscbc ^8^	file

^1^ sc: secret communication, ^2^ cac: confidentiality and certification, ^3^ std: spatial and transform domain, ^4^ m-b: modle-based, ^5^ avi: audio, video and image, ^6^ dp: data protection, ^7^ dian: data integrity, authentication and non-repudiation, ^8^ trscbc: transposition, replacement, stream cipher, block cipher.

**Table 2 sensors-21-01032-t002:** Comparison of existing steganography and steganalysis overviews based on VoIP.

Related Work	SCM ^1^	SCCEP ^2^	SSCEP ^3^	PSM ^4^	CCSM ^5^	PCSS ^6^	CPR ^7^
[15]	× ^8^	×	- ^9^	×	×	-	×
[16]	×	×	×	√ ^10^	√	×	×
[17]	×	√	×	√	√	×	×
[18]	×	×	√	×	×	√	×
Ours	√	√	√	√	√	√	√

^1^ SCM: Steganography and Confrontation Model, ^2^ SCCEP: Steganography classification based on embedding position, ^3^ SSCEP: Steganalysis classification based on embedding position, ^4^ PSM: Protocol stack model, ^5^ CCSM: Covert communication scene model, ^6^ PCSS: Performance comparison of steganalysis, ^7^ CPR: Compare with the previous reviews, ^8^ ×: no, ^9^ -: not involve, ^10^ √: yes.

**Table 3 sensors-21-01032-t003:** Evaluation coefficients in steganography.

Evaluation Indicators	1	2	3	4	5
Imperceptibility	PESQ	MOS ^1^	SNR	PSNR ^2^	Spectrogram
Hidden capacity	BPF	BPS	BPCoB ^3^	-	-
Robustness	TER	ADR	N-P T ^4^	-	-

^1^ MOS: Mean Opinion Score, ^2^ PSNR: Peak Signal-to-Noise Ratio, ^3^ BPCoB: bits per cluster of bits, ^4^ N-P T: non-parametric test.

**Table 4 sensors-21-01032-t004:** Performance valuation of steganography methods based on FCB.

Work	Technique	Performance Improved		
		Imperceptibility	Hidden Capacity	Robustness
[27]	MEPR ^1^	√		
[28]	NID ^2^		√	√
[29]	EPT-AP ^3^	√		
[25]	AFAS ^4^	√		√
[32]	PDM ^5^	√		√

^1^ MEPR: Minimum effective pulse replacement, ^2^ NID: codebook segmentation method based on Neighbor Segmentation, ^3^ EPT-AP: Encoding position transposition of adjacent pulses, ^4^ AFAS: AMR (Adaptive Multi-Rate) FCB (Fixed CodeBook) Adaptive steganography scheme, ^5^ PDM: Pulse distribution model.

**Table 5 sensors-21-01032-t005:** Summary of steganography methods on LPC.

Work	Technique	Performance Improved		
		Imperceptibility	Hidden Capacity	Robustness
[33]	MC-CH ^1^	√		
[37]	HOOK ^2^		√	√
[38]	ME ^3^	√		
[40]	CD-STC ^4^	√		√
[41]	NPP-QIM ^5^	√		√
[42]	PDM ^6^	√		√
[43]	OAET ^7^	√		√
[44]	B-G ^8^		√	

^1^ MC-CH: Matrix coding and chaos theory ^2^ HOOK: HOOK mechanism, ^3^ ME: matrix embedding technique, ^4^ MCD-STC: cepstrum distortion cost function and Syndrome Trellis Codes, ^5^ NPP-QIM: quantization-index-modulation based on the replacement of the nearest-neighbor projection point, ^6^ PDM: pulse distribution model, ^7^ OAET: optimized audio embedding technique. ^8^ B-G: bit-grading.

**Table 6 sensors-21-01032-t006:** Steganography methods based on ACB.

Work	Technique	Performance Improved		
		Imperceptibility	Hidden Capacity	Robustness
[46]	CS-ACB ^1^	√		
[48]	DLS ^2^		√	√
[50]	HideF0+ ^3^			√
[51]	PDAS ^4^	√		√
[19]	FPD-APMS ^5^		√	√
[53]	MME ^6^	√	√	

^1^ CS-ACB: control the scope of the ACB search, ^2^ DLS: double layer steganography, ^3^ HideF0+: an improved version of the HideF0, ^4^ PDAS: adaptive steganography algorithm based on the unvoiced pitch delay jitter characteristics, ^5^ FPD-APMS: adaptive partial matching steganography based on fractional pitch delay search, ^6^ MME: multi-matrix embedding.

**Table 7 sensors-21-01032-t007:** Steganography methods based on protocol.

Steganography Layer	Work	Technique	Performance Improved		
			Imperceptibility	Hidden Capacity	Robustness
Internet Layer	[56]	FSR ^1^ and SHS ^2^	√	√	
Transport Layer	[59]	Transcoding		√	√
	[60]	GM ^3^ and FIM ^4^			√
	[61]	StegUDP ^5^			√
	[63]	StegVAD ^6^		√	
	[64]	Hash			√
Application Layer	[65]	PHSL ^7^	√		√
	[67]	DC-FTP ^8^		√	

^1^ FSR: fast-starting retransmission, ^2^ SHS: spatial hiding synchronization, ^3^ GM: Gilbert model, ^4^ FIM: fractal interpolation model, ^5^ StegUDP: Steganography based on UDP packet length, ^6^ StegVAD: Steganography with Voice Activity Detection, ^7^ PHSL: Packet hierarchical sequence length covert channel, ^8^ DC-FTP: Directory coding for file transfer protocol.

**Table 8 sensors-21-01032-t008:** Evaluation coefficients in steganalysis.

Evaluation Indicators	1	2	3	4	5
Accuracy	ACC	FPR	FNR	TPR ^1^	TNR ^2^
Applicability	AC	ASA	OoO ^3^	AFF ^4^	-
Complexity	SC	TC	MS ^5^	MC ^6^	SHC ^7^

^1^ TPR: True Positive Rate, ^2^ TNR: True Negative Rate, ^3^ OoO: online or offline, ^4^ AFF: Applicable file format, ^5^ MS: Model structure, ^6^ MC: Memory consumption, ^7^ SHC: Software and hardware costs.

**Table 9 sensors-21-01032-t009:** Performance valuation of steganalysis methods based on FCB.

Work	Technique	Performance Improved		
		Accuracy	Applicability	Complexity
[69]	M-E ^1^	√		
[72]	F-SPPF ^2^	√	√	
[74]	ASOC ^3^	√	√	√
[75]	DST ^4^	√		√
[76]	SRCNet ^5^	√		
[77]	M-C F ^6^	√		
[78]	XGBoost ^7^	√		

^1^ M-E: Markov and Entropy, ^2^ F-SPPF: fast same pulse position feature, ^3^ ASOC: analysis of distribution characteristics, ^4^ DST: Dempster-Shafe Theory, ^5^ SRCNet: steganalysis combining RNN and CNN, ^6^ M-C F: multi-classifier fusion, ^7^ XGBoost: extreme gradient boosting.

**Table 10 sensors-21-01032-t010:** Steganalysis methods based on LPC.

Work	Technique	Performance Improved		
		Accuracy	Applicability	Complexity
[79]	PVS-PST ^1^	√		√
[47]	QCCN ^2^	√	√	
[80]	CCM-RNN ^3^	√		
[81]	K-fold CV ^4^	√		
[82]	CBNet ^5^		√	√
[83]	PCA ^6^	√		
[84]	FS ^7^			√
[85]	CTHC ^8^	√		

^1^ PVS-PST: phoneme vector space and phoneme state transition model, ^2^ QCCN: quantization codeword correlation network, ^3^ CCM-RNN: codeword correlation model based on RNN, ^4^ K-fold CV: K-fold cross-validation, ^5^ CBNet: Code Bayesian Network, ^6^ PCA: principal component analysis, ^7^ FS: fast steganalysis, ^8^ CTHC: calibration technology and hybrid classifier.

**Table 11 sensors-21-01032-t011:** Summary of steganalysis methods based on ACB.

Work	Technique	Performance Improved		
		Accuracy	Applicability	Complexity
[86]	CCN ^1^	√		√
[45]	C-MSDPD ^2^	√		
[89]	MTP ^3^	√		
[92]	SOD-OES ^4^	√	√	
[93]	SFFN ^5^	√		
[94]	BFSF ^6^	√		

^1^ CAN: codebook correlation network model, ^2^ C-MSDPD: calibrated Markov transition probability matrix of the second-order difference of pitch delay, ^3^ MTP: Markov transition probability, ^4^ SOD-OES: Second order difference and Odd even statistic, ^5^ SFFN: Steganalysis Feature Fusion Network, ^6^ BFSF: basic frequency statistical features.

**Table 12 sensors-21-01032-t012:** Steganalysis methods based on protocol.

Steganography Layer	Work	Technique	Performance Improved		
			Accuracy	Applicability	Complexity
Internet Layer	[95]	SVM ^1^	√		√
Transport Layer	[96]	CT ^2^ and PSR ^3^			√
	[97]	GMM ^4^ and MFCC ^5^	√	√	
Application Layer	[98]	CT ^2^			√
	[100]	M-F C ^6^	√		
	[101]	M-F C ^6^	√		

^1^ SVM: support vector machine, ^2^ CT: Chaos Theory, ^3^ PSR: phase space reconstruction technique, ^4^ GMM: Gaussian mixture model, ^5^ MFCC: Mel-frequency cepstral coefficients, ^6^ M-F C: multi-feature classification.

## Data Availability

Not applicable.

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
