# Peer review of "Steganography and Steganalysis in Voice over IP: A Review"

_sensors, 2021, doi:10.3390/s21041032_

Round 1
Reviewer 1 Report
Comments
1-Very poor 2 paragraph. Ie must lay the groundwork for the unfolding of the whole manuscript. First, the authors have to explain the base methods of steganalysys.
2-I can't accept tables without added numbers. How the authors define the terms as аccuracy, applicability, complexity, imperceptibility, hidden capacity, and robustness. Explain them. Without numbers the tables are meaningless.
Reviewer 2 Report
The authors review Steganography and Steganalysis in Voice over IP. Their contributions to the field are: The steganography and steganalysis algorithms are classified and summarized in detail according to the embedding area and parameter attributes simultaneously; Analyze the working principle of the VoIP steganographic transmission confrontation model; The existing articles are classified and summarized, and the performance of steganography algorithms are compared in terms of imperceptibility, hidden capacity, and robustness. Compare the performance of steganalysis algorithms with accuracy, applicability, and algorithm complexity; Summarized newer steganography and steganalysis methods, and proposed future development directions based on existing methods and challenges. The paper is very well written and organized. My suggestion is that the authors enlarge figures 2, 3, 4, and 5.
Reviewer 3 Report
I have reviewed the manuscript with ID: sensors-1059221 , the authors present the article entitled “ Steganography and Steganalysis in Voice over IP: A Review ”, which may be of interest to the community of this Journal, however, some aspects must be improved. Below I list some points to improve the quality of the manuscript and clarify its main contributions:
P1. Abstract. The authors present a brief background and extensive concepts about the topic. It is recommended to review in detail the Instructions of authors of this Journal, this in order to write the Reviwe article according to the quality standards of this Journal. For Review articles, the Abstract must provides concise and precise updates on the latest progress made in a given area of research. Systematic Reviews should follow the PRISMA. Also Provides a structured summary including, as applicable: background; objectives; data sources; study eligibility criteria, keywords, and interventions; study appraisal and synthesis methods; results; limitations; conclusions and implications of key findings; systematic review registration number.
PRISMA covers systematic reviews and meta-analyses. Authors are recommended to complete the checklist and flow diagram and include it with their submission.
https://www.mdpi.com/editorial_process#standards
http://prisma-statement.org/PRISMAStatement/Checklist
http://prisma-statement.org/PRISMAStatement/FlowDiagram
for more details please see:
Moher D, Liberati A, Tetzlaff J, Altman DG, The PRISMA Group (2009). Preferred Reporting Items for Systematic Reviews and Meta- Analyses: The PRISMA Statement. PLoS Med 6(7): e1000097.
DOI:10.1371/journal.pmed1000097
P2. Introduction. In lines 34 to 37, the authors say… Different from encryption technology, steganography technology provides a more reliable and safe method of embedded information to make the information undetectable for the third party [Please cite some references to support these arguments]. Please review the following papers: (you can review other papers)
DOI: 10.1109/TSMC.2019.2903785
DOI: 10.1007/s11042-020-08971-x
P3. Line 40. The authors say… In short, encryption technology hides the content of covert communication [Please cite the following refs. to support these arguments].
DOI: 10.1007/s11042-011-0956-1
DOI: 10.1016/j.chaos.2020.109646
P4. Line 42. the authors say… so steganography technology can provide better concealment and security [Please cite some references to support these arguments]. Please review the following papers: (you can review other papers)
DOI: 10.1109/ACCESS.2020.2971528
DOI: 10.1007/s11042-019-7559-7
P5. It would be interesting if the authors add more differences, characteristics and advantages versus cryptography, including chaotic encryption.
P6. Introduction. Line 86, the authors say… As we all know, there have been several reviews on VoIP steganography and steganal-ysis…. Please cite the references to support these arguments.
P7. Table 1. It says Work…. please complete by Related work
P8. Line 98. Please replace “This article”…. by The aim of this paper is…
P9. Line 162-164, the authors say… Steganalysis is a confrontation technology of steganography. Its target is to discover the presence of secret information and even damage confidential communication. Steganalysis is a vital echnology to resolve the issue of criminal use of steganography… please put [Reference
P10. Line 171. the authors say… The commonly used evaluation indicators of steganalysis include accuracy, applicability, and complexity [please cite some references to support these arguments]
P11. Line 185. After the discussion in the first two chapters…. replace chapters by sections.
P12. Line 193-194. the authors say… The steganography methods based on the voice payload has better imperceptibility and larger hiding capacity [please cite some references to support these arguments]
P13. For section 2, 3 and 4. Systematic Reviews should follow the PRISMA, i.e., the following is required: Indicate if a review protocol exists, Eligibility criteria, Information sources, Search, keywords, Study selection, Data collection process, Summary measures and Synthesis of results. Please review the check list.
http://prisma-statement.org/PRISMAStatement/Checklist
P14. Some important papers of the state of the art are being ignored. It is suggested to do a deep search for related work (state of the art) on main scientific databases and editorials (MDPI, IEEE, Scopus/Elsevier, Clarivate Analytics, Springer). Please try to review and cite at least 100 papers.
P15. Section 4. Steganalysis based on VoIP. In this section it is necessary to add more details of the main stego analysis techniques, for example, for accuracy, include the levels of accuracy reported in the main articles of the state of the art (not just put a check mark). Please do the same for Applicability, Complexity. It would be interesting if in the stego-analysis they add other parameters, such as: Peak Signal to Noise Ratio (PSNR), Invisibility (Perceptual Transparency), Hiding Capacity, Robustness, Security, among other.
P16. Conclusions are weak. Please improve the Conclusions, adding the key findings in this Review.
I hope that these comments and suggestions help to improve the quality of this paper.
Reviewer 4 Report
- What is the difference between signs “-“ and “x” in Table 1? Moreover, this table is not explained.
- Figures 3 and 6 are not explained. What are Ga and Gb?
- It should be added more information about VoIP. The authors describe FCB, ACB and LPC too brief. These techniques should be described in more detail and accompanied by some numerical examples, then corresponding part of the review will become much better.
- Analysis in this review is very weak. In Section 3 the authors just describe some steganography algorithms without discussion. The authors should compare different techniques of steganography and explain their advantages and disadvantages. By example, which of steganography techniques based on voice payload provides more robustness? More capacity?
- Section 4 has the same issue. The authors should improve the comparison of steganalysis techniques.
Round 2
Reviewer 1 Report
All of my comments are fixed. Thanks to the authors. The article has an improved structure.
I have no other comments. My decision is:
ACCEPT AS IS.
Author Response
Dear reviewer,
Thanks very much for taking your time to review this manuscript. We appreciate your comments and suggestions.
Thank you very much for your constructive comments on our work, and thank you for your recognition of our work.
Kind regards
Junjun Guo
Reviewer 3 Report
Thanks for address the suggestions, the manuscript looks well organized and clear.
Author Response

(The authors gave the same response as above.)

Reviewer 4 Report
The manuscript looks better now. However, in my view, the authors should add their responses on my comments 4,5 to the manuscript. By example, it may be a new table with review of different metrics which authors mention in their responses.
Author Response
Please see the attachment.

This manuscript is a resubmission of an earlier submission. The following is a list of the peer review reports and author responses from that submission.